# Fecal Microbiota Transplantation in NAFLD Treatment

**DOI:** 10.3390/medicina58111559

**Published:** 2022-10-30

**Authors:** Ludovico Abenavoli, Valentina Maurizi, Emanuele Rinninella, Jan Tack, Arianna Di Berardino, Pierangelo Santori, Carlo Rasetti, Anna Caterina Procopio, Luigi Boccuto, Emidio Scarpellini

**Affiliations:** 1Department of Health Sciences, University “Magna Græcia”, 88100 Catanzaro, Italy; 2Internal Medicine Residency Program, Università Politecnica delle Marche, 60121 Ancona, Italy; 3Clinical Nutrition Unit, Fondazione Policlinico A. Gemelli IRCCS, 00168 Rome, Italy; 4Department of Translational Medicine and Surgery, Università Cattolica del Sacro Cuore, 00168 Rome, Italy; 5T.A.R.G.I.D., Gasthuisberg University Hospital, KU Leuven, Herestraat 49, 3000 Leuven, Belgium; 6Clinical Nutrition and Internal Medicine Unit, “Madonna del Soccorso” General Hospital, 63074 San Benedetto del Tronto, Italy; 7Hepatology and Internal Medicine Unit, “Madonna del Soccorso” General Hospital, 63074 San Benedetto del Tronto, Italy; 8Healthcare Genetics and Genomics Doctoral Program, School of Nursing, College of Behavioral, Social and Health Sciences, Clemson University, 105 Sikes Hall, Clemson, SC 29631, USA

**Keywords:** gut microbiota, eubiosis, dysbiosis, bile acids, NAFLD, FMT

## Abstract

*Introduction*: Gut microbiota is not only a taxonomic biologic ecosystem but is also involved in human intestinal and extra-intestinal functions such as immune system modulation, nutrient absorption and digestion, as well as metabolism regulation. The latter is strictly linked to non-alcoholic fatty liver disease (NAFLD) pathophysiology. *Materials and methods*: We reviewed the literature on the definition of gut microbiota, the concepts of “dysbiosis” and “eubiosis”, their role in NAFLD pathogenesis, and the data on fecal microbiota transplantation (FMT) in these patients. We consulted the main medical databases using the following keywords, acronyms, and their associations: gut microbiota, eubiosis, dysbiosis, bile acids, NAFLD, and FMT. *Results*: Gut microbiota qualitative and quantitative composition is different in healthy subjects vs. NALFD patients. This dysbiosis is associated with and involved in NAFLD pathogenesis and evolution to non-acoholic steatohepatitis (NASH), liver cirrhosis, and hepatocellular carcinoma (HCC). In detail, microbial-driven metabolism of bile acids (BAs) and interaction with hepatic and intestinal farnesoid nuclear X receptor (FXR) have shown a determinant role in liver fat deposition and the development of fibrosis. Over the use of pre- or probiotics, FMT has shown preclinical and initial clinical promising results in NAFLD treatment through re-modulation of microbial dysbiosis. *Conclusions*: Promising clinical data support a larger investigation of gut microbiota dysbiosis reversion through FMT in NAFLD using randomized clinical trials to design precision-medicine treatments for these patients at different disease stages.

## 1. Introduction

Gut microbiota can be defined as a complex ecosystem encompassing more than 100 trillion microbes belonging to bacteria, viruses, archaea, protozoa, fungi, and yeasts living in our intestines. However, this taxonomic definition does not describe its functioning, which strictly relies on qualitative and quantitative composition [1]. When the qualitative and quantitative composition of gut microbiota remains stable we can speak about “eubiosis” [1]. When this composition is altered by intestinal and environmental factors (e.g., use of antibiotics, or gastrointestinal and extraintestinal infections), “dysbiosis” is established [2]. There are different dysbioses for different gastrointestinal and extra-intestinal diseases [3,4]. Thus, this significant association is gaining more and more attention from researchers as it can help the development of “personalized” medicine [5]. Non-alcoholic fatty liver disease (NAFLD) is a complex condition characterized by fat deposits within hepatocytes [6]. Liver fat deposition is significantly associated with obesity, dyslipidemia, type 2 diabetes, and insulin resistance as comorbidities [7,8]. Almost one-quarter of the Westernized population is affected by this condition which has a spectrum of stages [9]. Aside from fat deposition, about 10–15% of patients can develop liver fibrosis because of progressive liver inflammation [10]. Liver fibrosis characterizes non-alcoholic steatohepatitis (NASH). If liver fibrosis is not reverted and/or stopped by treatments and lifestyle changes, patients can develop liver cirrhosis as well [11]. Particularly, NAFLD patients can show hepatocellular carcinoma (HCC) also without liver cirrhosis at any stage [12]. In NAFLD, gut microbiota derangement, namely dysbiosis, is resembled by an increased abundance of *Bacteroidetes* and a decrease of those of *Firmicutes* [13]. This starts a pathophysiological process through the “gut-liver axis “ where antigens from intestinal pathogens enter the portal blood cycle via a deranged intestinal permeability through damage-associated molecular patterns (DAMPS) and pathogen-associated molecular patterns (PAMPS) [14]. This pathophysiological cascade leads the liver to be exposed to detrimental antigens that start and perpetuate liver damage [14]. Conversely, liver inflammation favors altered intestinal permeability increase [14]. Bile acids (BA) are involved in lipid absorption and metabolism regulation [15]. Gut microbiota is also implicated in such regulation and actively interacts with the free BAs pool and their receptors, particularly the farnesoid X receptor (FXR), present both in enterocytes and hepatocytes, with different and tissue-specific effects on liver fat deposition and fibrosis [16]. The use of antibiotics, pre-, and probiotics to restore gut “eubiosis” in NAFLD patients has shown limitations as well as promising results. Probiotics can protect the liver from toxic substances driven to the liver by altered intestinal permeability. In addition, they can restore altered intestinal permeability, and gut eubiosis and exert a direct anti-inflammatory effect on gut and liver tissue. Antibiotics can restore gut eubiosis and prevent the pathophysiologic cascade characteristic of the “gut-liver axis”. Unfortunately, probiotics efficacy needs larger randomized clincal trials (RCT)s investigating their safety. Antibiotics suffer from the emerging issue of antibiotic resistance development within the gut, with potential deleterious side effects for the whole human body [17]. Direct fecal microbiota transplantation (FMT) is a safe and very successful treatment for GI infections such as those by *Clostridium difficile* and other *Clostridiales* [18]. In particular, *Clostridium difficile* infection (CDI) is a worldwide healthcare problem as it is associated with severe diarrhea until toxic megacolon occurrence. CDI is typically associated with the use of antibiotics, and recurrent CDI (rCDI) has significant mortality. FMT has been seen to confer a high success rate in the treatment of CDI and rCDI as it restores the commensal gut microbiota and re-establish “indirect mechanisms“ of colonization resistance. Indeed, FMT restores microbiome-mediated regulation of the integrity of the gut barrier function to prevent penetration/translocation of potential pathogens (namely, *Clostridiales*). Several preclinical and some clinical evidence support the use of FMT also in NAFLD patients [19]. However, the role of FMT as a therapeutic choice in NAFLD has not been fully elucidated. Thus, we first review the literature on gut microbiota physiologic composition, the concepts of “dysbiosis” and “eubiosis”, and their role in the pathogenesis and pathophysiology of NAFLD. Finally, we reviewed the data on FMT use in preclinical and initial human studies.

## 2. Materials and Methods

We conducted a PubMed and Medline search for original articles, reviews, meta-analyses, and case series using the following keywords, their acronyms, and associations: «gut microbiota», «eubiosis», «dysbiosis», «bile acids», «NAFLD», «FMT». When appropriate, preliminary evidence from abstracts belonging to main national and international gastroenterological meetings (e.g., United European Gastroenterology Week, Digestive Disease Week) was also included. The papers found from the above-mentioned sources were reviewed by two of the authors (ES and ER) according to PRISMA guidelines [20]. The last MEDLINE search was dated 31 July 2022.

## 3. Results

### 3.1. Gut Microbiota Composition in Health and Its “dysbiosis” in NAFLD Patients

The human intestine hosts over 100 trillion microbes, prevalently bacteria. There are also viruses, fungi, archaea, and protozoa [21]. Gut microbiota participates in nutrient absorption and fermentation, modulates intestinal permeability (IP), and is implicated in host metabolism (e.g., carbohydrates absorption and processing, proteins putrefaction, bile acids formation, insulin sensitivity), and modulation of mucosal and systemic immunity [22]. Firmicutes and Bacteroidetes represent the most abundant phyla of the bacterial microbiota. Their ratio is a fundamental factor in the host’s health [23]. Among the other phyla present, we find Fusobacteria, Actinobacteria, Proteobacteria, and Verrucomicrobia [24]. Eubiosis defines the balanced qualitative and quantitative condition of the intestinal microflora and is essential to preserving the host’s health [22,23,24]. On the contrary, its qualitative/quantitative perturbation–dysbiosis is associated with the development of various diseases such as NAFLD, NASH until HCC, diabetes type 2, cardiovascular disorders, and, undoubtedly, obesity [25,26,27,28]. Patients with NAFLD, during their stages of disease evolution, also have an increased number of Bacteroidetes and a reduction of Firmicutes abundance (a decreased F/B ratio) [29]. However, the F/B ratio may vary across studies mainly because of the gut microbiota studying technique. There are available culture-based and culture-free methods such as 16S rRNA gene sequencing until whole-genome shotgun sequencing. Finally, there are newer metagenomics approaches that allow whole genomic sequence reconstruction starting from a few genetic fragments present in the biological sample in the study. Interestingly, there are diversities in the microorganisms’ abundance within each phylum. NAFLD patients also show an increased proportion of species belonging to *Clostridium*, *Anaerobacter*, *Streptococcus*, *Escherichia*, and *Lactobacillus*, whereas *Oscillibacter*, *Flavonifractor*, *Odoribacter*, *Alistipes* spp. are less abundant [30]. More interestingly, the hallmark of decreased F/B ratio is not typical of patients suffering from HCC. Of interest, in children with NAFLD and NASH, there is a peculiar gut dysbiosis characterized by decreased abundance of *Oscillospira* spp. and by an increased concentration of *Dorea*, *Blautia*, *Prevotella copri*, and *Ruminococcus* spp. vs. in healthy children [31]. In terms of diversity, adult NAFLD gut microbiota shows decreased α-diversity (richness and evenness), significantly altered β-diversity, and significant differences in the abundance of bacteria at the phylum, class, family, or genus level vs. healthy subjects. Interestingly, NAFLD children show decreased α-diversity, distinct differences in β-diversity, or differing abundance of bacteria at the phylum or genus levels vs. healthy subjects. 

It is worth mentioning the role of other factors in maintaining the dynamic balance of gut eubiosis in the every-day life of humans: diet, use of antibiotics, aging, diseases affecting GI nutrients absorption, motility, and inflammatory state. For example, animal experiments have shown how a high-fat diet is able to give an “obese shape” to our gut microbiota. On the other hand “obese gut microbiota transplantation” is able to change gut microbiota behavior in lean mice [32]. 

Diet, and related lifestyle, have a particular impact on NAFLD pathogenesis and physiopathologic evolution. However, the gut microbiota is a peculiar actor in NAFLD’s natural history that interacts with other specific actors within the gut, leading to specific liver steatosis and fibrosis features [6] (Figure 1).

### 3.2. Bile Acids (BA), Gut Microbiota, and NAFLD

The first physiologic interaction between BAs and microbiota happens in the large intestine where the microorganisms carry out several enzymatic reactions (e.g., deconjugation, dihydroxylation, and epimerization) to form secondary BAs such as deoxycholic acid (DCA) from cholic acid, and ursodeoxycholic acid (UDCA) and lithocholic acid (LCA) from chenodeoxycholic acid (CDCA) [33,34]. Indeed, BAs deconjugation is crucial to make BAs more lipophilic. Secondary BAs can be reabsorbed in the large bowel and return to the liver, via systemic circulation, allowing their “entero-hepatic circle” [23,35]. One proof of the strict relationship and evolutive interaction between intestinal microbiota and BAs is the presence of BAs hydrolases (BSH) in these microbes. In particular, BSH is present in most bacterial phyla; in Gram-positive bacteria, like Lactobacillus, Bifidobacterium, Clostridium spp., and Enterococcus, and in some commensal Gram-negative strains such as the Bacteroides spp. [36,37,38,39]. In pathogens such as Listeria monocytogenes, BSH activity probably represents an adaptive quality guaranteeing its gut persistence [40]. Interestingly, this adaptive quality has a horizontal transmission amongst bacteria [41]. More interestingly, the human gut can affect BSH activity in gut bacteria through a “host species-specific selection” of microbial BSH activities according to “species-specific” differences in BAs pools [42]. Indeed, BSH activity is a “protective shield” for bacteria colonizing the human gut [43]. In the frame of NAFLD, the consolidated concept of the gut–liver axis helps the understanding of the multi-directional interaction of BAs, gut microbiota, and liver “through” the intestine [14]. BAs are ligands for several receptors. They include FXR and G-protein-coupled bile acid receptor 1 (or TGR5). They mainly modulate host metabolism and BAs enterohepatic circulation [44]. BAs are natural antibacterial substances and maintain gut microbial eubiosis through the activation of FXR [45]. Physiologically, hepatic FXR activation by BAs induces the expression of atypical nuclear receptors small heterodimer partner (SHP), which promotes the inhibition of the sterol-regulatory element-binding protein-1c (SREBP-1c) with a reduced hepatic synthesis of triglycerides. In addition, FXR physiologically limits fat accumulation in the liver [46,47]. Moreover, FXR activation in the liver results in the inhibition of gluconeogenesis and glycolysis. These two effects have a potential protective role in insulin resistance and type II diabetes [46,47]. Interestingly, in NAFLD patients the BAs-FXR interaction is altered such as the resulting lipid and glucose metabolism [16]. Gut microbiota derangements are dynamically affected by BAs pool changes, and, conversely, BAs pool composition and abundance are affected by gut microbiota eubiosis and dysbiosis. In NAFLD patients, gut dysbiosis can alter the BAs pool and start a pathophysiological cascade that favors liver fat accumulation and microinflammation, which can result in fibrosis. The latter is a typical feature of NASH [48] (Figure 1). Bariatric surgery patients offer an example of FXR regulation by bile acids. In bariatric surgery on obese patients, the metabolic improvements obtained upon the procedure follow both weight-dependent and weight-independent mechanisms. Substantially, increased volumes of digestate-free BAs are delivered to the distal gut, with increased hepatic flux/enterohepatic circulation, increased plasma BA levels and composition, and, subsequent normalization of the blunted postprandial plasma BA concentration typical of obesity. Altogether, this results in modulation of satiety, improvement of lipid and cholesterol metabolism, incretins and glucose homeostasis, and energy storage and distribution via fine FXR and TGR5 interaction of BAs. 

### 3.3. From Probiotics Use to Fecal Microbiota Transplantation (FMT) in NALFD Treatment 

Several pieces of evidence from both in vitro and in vivo studies have shown that probiotics—“alive organisms beneficially affecting host (human) health”—can compete with pathogenic bacteria, improve intestinal permeability to pathogens and/or their molecules, exert immune-modulatory effects and gut-brain axis effects with the production of neurotransmitters [49]. In detail, probiotics can significantly decrease endotoxin levels within the gut [50]. Subsequently, the first human studies have revealed how probiotics can also clinically ameliorate NAFLD fat deposition, micro-inflammation, and fibrosis onset and progression. In detail, in overweight NAFLD patients, administration of multistrain probiotic mixtures reduces the systemic inflammatory state and, more interestingly, brings back the gut microbiota towards a “commensal” one [51,52]. However, these promising shreds of evidence have to face a solid Cochrane meta-analysis from 2007 that did not come to a clear conclusion substantially because of the lack of a sufficient number of randomized controlled trials [53]. Thus, RCTs have been doubling in the few last years. The multistrain VSL#3® and a combination of pro/prebiotics as symbiotics were able to reduce levels of TNF-α, liver transaminases, and oxidative stress markers in NAFLD patients for a duration of 2- 3 months [54]. Similar results have been obtained in children [55,56,57,58]. (Table 1). 

In more detail, multistrain preparations show a significant gut microbiota reshaping in both NAFLD and NASH patients. In addition, their effect on liver function and biochemical lipid profile is positive: several RCTs report liver transaminases reduction with cholesterol and triglycerides reduction. There is also a significant reduction in inflammatory cytokines. It is important to mention that most of the enrolled patients were also type 2 diabetes subjects (Table 1).

However, dose- and strain(s)-finding studies are still missing and data on adverse effects linked to the use of probiotics in NAFLD patients. Thus, the use of live commensals coming directly from a healthy gut is appealing and may (in theory) guarantee a safer use than probiotics. These issues have paved the road for use of FMT in NAFLD patients. Several animal studies offered the first evidence of FMT in NAFLD. Leroy et al. showed that NAFLD- mice are able to induce the development of NAFLD in the vast majority of recipients of FMT. This finding was confirmed by the species-specific rise of bacteria being predominant colonizers upon FMT [65]. Furthermore, Zhou et al. studied two groups of high-fat diet (HFD)-fed mice; one of those underwent FMT from healthy donors. This group showed a significant reduction in histological findings typical of NAFLD (namely, intracellular hepatic lipid and proinflammatory cytokines concentration) vs. those not receiving healthy FMT [66]. More systematically, FMT has been successfully used in HFD-NAFLD mice. Interestingly, also NASH histological findings (namely, liver fibrosis and inflammatory infiltrates) have been demonstrated to improve after FMT. This healing was correlated with body weight, fat content, and serum reduction of transaminases levels [67]. Anecdotal reports show a fine interaction between gut microbiota and NAFLD in humans, including eighteen patients with metabolic syndrome diagnosis received either self-FMT or from lean healthy subjects. Interestingly, six weeks after FMT, insulin sensitivity significantly improved, and this improvement was correlated with an increase in butyrate-producing bacteria [68]. Phillips et al. confirmed the efficacy of FMT in the treatment of alcoholic hepatitis patients, also describing the gut dysbiosis changes after treatment [69]. In this pilot study, eight male patients with severe acute alcoholic hepatitis (SAH) consented to the FMT procedure. FMT consisted of thirty grams of donor stool samples, obtained after screening from consenting family members, homogenized with 100 mL of sterile normal saline, and filtered using sterile gauze. Therefore, small amounts were infused through a nasoduodenal tube daily for one week. Control subjects were historical patients with SAH under standard-of-care treatment. Samples for the gut microbiota study were collected at baseline, 6-month, and one-year intervals. Interestingly, one week of FMT was able to improve liver disease severity indexes and, more importantly, the survival rate at 1 year after treatment. In addition, FMT was confirmed to be a safe procedure, recording just flatulence as a mild side effect. In detail, and more intriguingly, there was the coexistence of donor and recipient gut microbial species at 6 and 12 months after FMT in treated patients. This finding describes a peculiar behavior of transplanted microbial species from donors: they are less pathogenic and beneficial and coexist with pre-existing bacterial communities instead of replacing those of the recipient. The latter are “beneficially” modified by bacteria from donors, resulting in a new-one “symbiotic coexistence”. This data finds initial agreement in the literature [70]. Very recently, a study by Xue et al. has better highlighted the different impacts of FMT on NAFLD according to the presence or absence of obesity [71]. NAFLD patients were randomized to receive either oral probiotics or FMT. FMT was prepared from donor stool (heterologous), administered through colonoscopy, and followed by three enemas over 3 days. Both groups were required to undergo healthy lifestyle changes (namely, a healthy diet and regular physical exercise for more than 40 min per day). Patients were rechecked 1 month after treatment (e.g., liver fat deposition, fecal microbiota composition). Interestingly, FMT was able to significantly decrease liver fat accumulation and reduce gut microbiota dysbiosis. Importantly, FMT had a significantly higher healing efficacy on lean NAFLD vs. obese patients. This finding was consensual with a more marked restoration of gut eubiosis in lean vs. obese patients. Indeed, this study reinforces evidence from the literature showing a different impact of gut dysbiosis in lean fatty liver, which is not correlated to impaired lipid and glucose metabolism issues such as obese fatty liver disease [72]. Currently, it remains to be investigated whether or not the FMT impact NAFLD and NASH complications. In detail, the RCTs available in the literature have shown a good efficacy of FMT used in NAFLD, NASH diabetic, and non-diabetic patients with improvement of glycemic control, and liver steatosis. However, some trials showed better efficacy in liver fibrosis reversal (NASH feature) vs. liver steatosis (namely, NAFLD subjects). Finally, some promising report confirms FMT safety also in liver cirrhosis patients, considered as an evolution of NASH (Table 2). 

Overall, FMT can be considered a therapeutic, safe treatment in NAFLD patients, and perhaps it is a step up from NASH.

## 4. Conclusions

Gut microbial dysbiosis is a validated pathogenic mechanism connected to the pathophysiology of NAFLD and its stages. The type and severity of dysbiosis differ between patients with NAFLD, NASH, cirrhotic, and HCC. In addition, it is worth mentioning the recent evidence describing the different weights of gut dysbiosis in lean vs. obese NAFLD patients. This evidence is mirrored by the major efficacy of amelioration of liver fat deposition and reduction of dysbiosis of FMT in lean vs. obese NAFLD patients. FMT is gaining more and more evidence for use as a treatment of NAFLD, as probiotics and lifestyle changes have several limits and risks for patients. Probiotics use should be “personalized” for different NAFLD patients; accurate dose-finding and adverse events studies are needed. Lifestyle changes show the best efficacy in the treatment of obese patients. However, this subset of patients shows compliance with short- and long-term difficulties in mining the use of this life asset over the entire lifespan. Thus, FMT seems to be a safe, efficient treatment for NAFLD patients. However, more RCT studies and long-term follow-ups are needed to verify their efficacy, especially in lean NAFLD patients, often not beneficially affected by lifestyle changes, cholesterol-lowering agents and probiotics use.

## Figures and Tables

**Figure 1 medicina-58-01559-f001:**
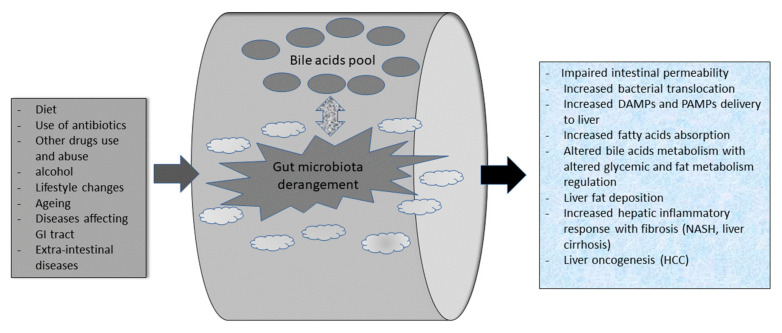
The complex interaction between factors affecting gut microbiota composition, its interaction with bile acids pool, and NAFLD pathogenesis. GI: gastrointestinal; DAMPs: damage associated microbial products; PAMPs: pathogens associated microbial products; NASH: non-alcoholic steatohepatitis.

**Table 1 medicina-58-01559-t001:** Clinical trials using probiotics for NAFLD patients’ treatment.

Patients	Type of Study	Probiotics Used	Outcomes	Ref
48 children with NAFLD	Randomized clinical trials (RCT)	VSL#3 supplementation for 4 months vs. placebo	NAFLD reversal	[57]
64 children with obesity and NAFLD	RCT (triple blind)	*L. acidophilus*, *B. lactis*, *B. bifidum*, *L. rhamnosus*	↓ALT, AST, mean cholesterol, LDL-C, and TG in the probiotic-administered group	[58]
38 patients (16 with NASH diagnosis): 7 treated with Probiotics, 9 with the standard of care vs. 22 healthy controls	RCT	*Lactobacillus* ssp. and *Bifidobacterium* *bifidum*	In NASH patients there was gut microbiota modulation: ↓Faecalibacterium ↓Anaerosporobacter ↑Parabacteroide ↑Allisonella	[59]
48 T2DM NAFLD patients	RCT	*“Symbiter Omega”* probiotic biomass vs. placebo	↓Fatty acids, ↓serum gamma-glutamyl transpeptidase ↓TG ↓TC	[60]
58 patients with NAFLD and T2DM: 30 treated with probiotics vs. 28 receiving a placebo	RCT	biomass of 14 probiotic bacterial genera	In NAFLD patients: ↓Liver fat deposition ↓aminotransferase ↓TNF-α and IL-6	[51]
200 patients with NAFLD randomized to control group (standard of care treatment) and add-on treatments groups A, B, and C.	RCT	*Bifidobacterium*, *Lactobacillus*, *Enterococcus*, *Bacillus subtilis*, and *Enterococcus*	Amelioration of fatty liver deposition, ↓ALT, AST and TNF-α ↑HMW-APN	[61]
75 patients with NASH under a low-fat/low-calorie diet	RCT	*Lactobacilli*, *Bifidobacteria*, *Streptococcus* *thermophilus*	Gut microbiota modulation towards “healthy” one, ↓BMI, ↓cholesterol in the probiotic-treated group	[62]
50 patients (42 NAFLD) were randomized to probiotic or placebo	RCT	*L. casei*, *L. acidophilus*, *L. rhamnosus*, *L. bulgaricus*, *B. breve, B. longum*, *S. thermophilus*	↓glycemic, inflammatory markers, Insulin, insulin resistance in NAFLD patients	[63]
30 patients with NAFLD	RCT	*L. bulgaricus* and *S. thermophilus*	↓ALT, AST, GGT in the probiotics group	[56]
72 patients with NAFLD	RCT	Probiotic yogurt	↓ALT, AST, TC, LDL-C	[64]

NAFLD: non-alcoholic fatty liver disease, NASH: non-alcoholic steatohepatitis, T2DM: type 2 diabetes mellitus, TNF-α tumor necrosis factor-α, IL-6: interleukin 6, BMI: body mass index, LDL: low-density lipoprotein, TC: total cholesterol, LDL-C: LDL cholesterol, HMW-APN high molecular weight adiponectin, LDL-C: low-density lipoprotein cholesterol, TG: triglycerides, GGT: gamma-glutamyl transferase, ALT alanine aminotransferase, AST aspartate aminotransferase, ↑ increase, ↓ decrease.

**Table 2 medicina-58-01559-t002:** Clinical trials using FMT for NAFLD treatment.

Study Type	Patients	Outcomes	RCT Number
FMT via nasojejunal tube	Diabetes and NAFLD	Improved HOMA index	NCT02469272
FMT, pilot study	NAFLD and NASH	Improved degree of liver steatosis as assessed by MRI	NCT02469272
FMT via duodenal infusion	NAFLD and NASH	Efficacy in NASH treatment vs. NAFLD	NCT03803540
FMT via duodenal infusion	NAFLD and NASH	Reduction of hepatic venous gradient pressure	NCT02721264
FMT vs. standard treatment (RCT)	Liver cirrhosis derived from NASH	Safety (e.g., number of adverse events, complications rate)	NCT02868164

NAFLD: non-alcoholic fatty liver disease; NASH: non-alcoholic steatohepatitis; FMT: fecal microbiota transplantation; HOMA: homeostasis model assessment; MRI: magnetic resonance imaging.

## Data Availability

All the data reported in this review of literature are available online on PubMed and main national and international gastroenterological meetings websites.

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
