# Peer review of "Fecal Microbiota Transplantation in NAFLD Treatment"

_medicina, 2022, doi:10.3390/medicina58111559_

Round 1

Reviewer 1 Report

Reviewer comments and suggestions

The authors in this study compared the gut microbiota qualitative and quantitative composition in healthy subjects vs. NALFD patients. They found that dysbiosis was associated and involved in NAFLD pathogenesis and evolution to steatohepatitis (NASH), liver cirrhosis, and hepatocellular carcinoma (HCC). The authors also discussed the bile acids (BAs) microbial-driven metabolism and interaction with hepatic and intestinal farnesoid nuclear X receptor (FXR) which may have role in liver fat deposition and development of fibrosis. In the end, the study concluded that clinical data support a larger investigation of gut microbiota dysbiosis reversion through FMT in NAFLD using randomized clinical trials to design precision-medicine treatments. 

The manuscript requires thorough proofreading from English native speakers as there were many flaws in terms of grammar and structural organization of sentences. Although the result was clear and concise. I listed below a few concerns and therefore it should be modified to gaining the quality of the manuscript

  1. The study needs a diagram that explains the relationship between gut microbiota and NAFLD.
  2. The authors need to add the success and failure of clinical trials that they discussed
  3. Line 54, type 2 not II, please change the format in your manuscript
  4. Line 71, with variegate effects on liver fat deposition and fibrosis what does it mean?
  5. Line 73 Explore these points, reference 17
  6. Line 75-76 The authors need to add more studies and explore it
  7. Line 107-108 Explore more 2-3 lines on the same topic
  8. Line 113-116 It would be nice if the authors could differentiate child with adults diversity in the microorganism
  9. Line 158-159 Explain the study here only, reference number 48

Author Response

REPLY POINT TO POINT TO REFEREES

Changes throughout the manuscript are marked in red.

#Referee 1

The authors in this study compared the gut microbiota qualitative and quantitative composition in healthy subjects vs. NALFD patients. They found that dysbiosis was associated and involved in NAFLD pathogenesis and evolution to steatohepatitis (NASH), liver cirrhosis, and hepatocellular carcinoma (HCC). The authors also discussed the bile acids (BAs) microbial-driven metabolism and interaction with hepatic and intestinal farnesoid nuclear X receptor (FXR) which may have role in liver fat deposition and development of fibrosis. In the end, the study concluded that clinical data support a larger investigation of gut microbiota dysbiosis reversion through FMT in NAFLD using randomized clinical trials to design precision-medicine treatments.

The manuscript requires thorough proofreading from English native speakers as there were many flaws in terms of grammar and structural organization of sentences. Although the result was clear and concise. I listed below a few concerns and therefore it should be modified to gaining the quality of the manuscript.

Reply: we thank the reviewer for this observation. We have done proofreading of the text according to the suggestion of the reviewer.

The study needs a diagram that explains the relationship between gut microbiota and NAFLD.

Reply: we thank the reviewer for this observation. We have added a figure describing the relationship between GM and NAFLD.

    The authors need to add the success and failure of clinical trials that they discussed.

Reply: we thank the reviewer for this observation. We added the required data, commenting on RCTs reported.

    Line 54, type 2 not II, please change the format in your manuscript

Reply: we thank the reviewer for this observation. We edited the text according to the suggestion of the reviewer.

    Line 71, with variegate effects on liver fat deposition and fibrosis what does it mean?

Reply: we thank the reviewer for this observation. We have made the sentence more understandable.

    Line 73 Explore these points, reference 17

Reply: we thank the reviewer for this observation. We have explored the highlighted points.

    Line 75-76 The authors need to add more studies and explore it.

Reply: we thank the reviewer for this observation. We have added more data on the evidences showing efficacy of FMT on Clostridiales infections.

    Line 107-108 Explore more 2-3 lines on the same topic

Reply: we thank the reviewer for this interesting observation. We have added some line on the topic.

    Line 113-116 It would be nice if the authors could differentiate child with adults diversity in the microorganism.

Reply: we thank the reviewer for this interesting observation. We have added the required data.

    Line 158-159 Explain the study here only, reference number 48.

Reply: we thank the reviewer for this interesting observation. We have developed the topic.

Reviewer 2 Report

This review evaluates the utility of fecal microbiota transplant (FMT) as a therapy for non-alcoholic fatty liver disease (NAFLD). Literature review indicates the microbial composition (quality and quantity) in NAFLD patients differs from that of controls and is associated with development and progression of disease. Further, liver fat deposition and fibrosis is related to microbial influences on bile acid metabolism and farnesoid X receptor (FXR) interaction. The literature indicates FMT utilization in NAFLD should be further explored.

Major:

1.       For Materials and Methods, what about the interactions of diet, comorbid conditions, age, etc. on microbiome and NAFLD development?

2.       Please add some discussion on the purported mechanism of microbiota-associated NAFLD. A figure here would be helpful also.

Minor:

1.       Abstract—Line 24. Instead of “physiopathology”, use “pathophysiology.” Line 25, delete “data.” Line 30, add “with” after “associated”. Lines 31-32, revise to “In detail, microbial-driven metabolism of bile acids (BAs) and interaction with…”.

2.       Line 48=49—revise to “dysbiosis is established.” Line 64—revise “circle” to “cycle.” Line 70, extra space.

3.       Line 194, fix “Anecdotical” to “Anecdotal.”

Author Response

#Referee 2

This review evaluates the utility of fecal microbiota transplant (FMT) as a therapy for non-alcoholic fatty liver disease (NAFLD). Literature review indicates the microbial composition (quality and quantity) in NAFLD patients differs from that of controls and is associated with development and progression of disease. Further, liver fat deposition and fibrosis is related to microbial influences on bile acid metabolism and farnesoid X receptor (FXR) interaction. The literature indicates FMT utilization in NAFLD should be further explored.

Major:

  1. For Materials and Methods, what about the interactions of diet, comorbid conditions, age, etc. on microbiome and NAFLD development?

Reply: we thank the reviewer for this observation. We have added a small section describing these factors interaction with microbiome and, NAFLD. It is not really the topic of the present contribution focused on gut microbiota and its transplantation in NAFLD treatment.

  1. Please add some discussion on the purported mechanism of microbiota-associated NAFLD. A figure here would be helpful also.

Reply: we thank the reviewer for this observation. We have added a figure describing the relationship between GM and NAFLD.

Minor:

  1. Abstract—Line 24. Instead of “physiopathology”, use “pathophysiology.” Line 25, delete “data.” Line 30, add “with” after “associated”. Lines 31-32, revise to “In detail, microbial-driven metabolism of bile acids (BAs) and interaction with…”.
  2. Line 48=49—revise to “dysbiosis is established.” Line 64—revise “circle” to “cycle.” Line 70, extra space.
  3. Line 194, fix “Anecdotical” to “Anecdotal.”

Reply: we thank the reviewer for these observations. We have addressed them accordingly.